# A Form of Non-Volatile Solid-like Hexadecane Found in Micron-Scale Silica Microtubule

**DOI:** 10.3390/ma16010009

**Published:** 2022-12-20

**Authors:** Weiqing An, Xiangan Yue, Jirui Zou, Lijuan Zhang, Yu-Chun Fu, Rongjie Yan

**Affiliations:** 1State Key Laboratory of Petroleum Resources and Prospecting, China University of Petroleum (Beijing), Beijing 102249, China; 2College of Petroleum Engineering, China University of Petroleum (Beijing), Beijing 102249, China; 3Key Laboratory of Petroleum Engineering Ministry of Education, China University of Petroleum (Beijing), Beijing 102249, China; 4Department of Chemical Engineering, National Chung Cheng University, Taiwan 621301, China

**Keywords:** solid-liquid interface, transmission electron microscope, silica, microtubules, hexadecane, surface-silanol

## Abstract

Anomalous solid-like liquids at the solid–liquid interface have been recently reported. The mechanistic factors contributing to these anomalous liquids and whether they can stably exist at high vacuum are interesting, yet unexplored, questions. In this paper, thin slices of silica tubes soaked in hexadecane were observed under a transmission electron microscope at room temperature. The H-spectrum of hexadecane in the microtubules was measured by nuclear magnetic resonance. On the interior surface of these silica tubes, 0.2–30 μm in inside diameter (ID), a layer (12–400 nm) of a type of non-volatile hexadecane was found with thickness inversely correlated with the tube ID. A sample of this anomalous hexadecane in microtubules 0.4 μm in ID was found to be formable by an ion beam. Compared with the nuclear magnetic resonance H-spectroscopy of conventional hexadecane, the characteristic peaks of this abnormal hexadecane were shifted to the high field with a broader characteristic peak, nuclear magnetic resonance hydrogen spectroscopy spectral features typical of that of solids. The surface density of these abnormal hexadecanes was found to be positively correlated with the silanol groups found on the interior silica microtubular surface. This positive correlation indicates that the high-density aggregation of silanol is an essential factor for forming the abnormal hexadecane reported in this paper.

## 1. Introduction 

Liquids on solid surfaces are involved in many surface reactions. These surface reactions include electrochemistry at liquid/solid interfaces [1,2,3], heterogeneous catalyses such as hydrogen production by catalytic reforming [4], rock weathering, and corrosion of metals [5]. The properties of the liquid at the solid interface also significantly impact the liquid’s heat transfer [6] and flow characteristics on the solid surface, especially in small-scale spaces—for example, micro-nano-scale pipe flow [7,8] and nonlinear seepage [9] in reservoirs, and other issues. Despite the importance of solid–liquid interfacial science, much anomalous behaviour of the liquids on solid surfaces is still far from being thoroughly researched, with many new phenomena yet to be discovered and understood.

Over the past two decades, most previous research has predominantly used molecular dynamics simulations to study the fluid phase behaviour at the liquid–solid interface with the vast majority of the subject liquid of study being water [10,11,12,13,14,15,16,17,18,19,20,21]. From these studies, the research has shown that certain structural characteristics and properties of the liquid on the solid surface undergo a change. For example, high-density water at the solid–liquid interface [9,10], ice-like water on the solid surface [11,12], ordered water at the solid–liquid interface [13,14,15], stable two-dimensional ice at room temperature [16], structured water and alkane layer at the solid interface [17,18], water with abnormal dielectric constant [19], and interfacial water and alkane with small self-diffusion coefficient [20,21].

Experimental research on the anomalous phenomena of the solid-fluids interface has also been reported. Klein et al. [22,23] used a surface force meter to measure the viscous forces of the water and hexadecane on mica, and found that its viscosity appeared to be solid-like for a thickness of several molecular layers. Crupi et al. [24] used incoherent quasi-elastic neutron scattering and Fourier infrared spectroscopy to characterize the water in the nano-scale pores of zeolite and found its characteristic spectrum anomalous as compared with bulk water. Anderson et al. [25] used Raman spectroscopy to study the water layer on the surface of the silica, and speculated the existence of an “ice-like” structure from the results. Algara-Siller et al. [26] observed water between two graphene sheets at room temperature and reported a “square ice” morphology under high-resolution electron microscopy. Takehiko et al. [27] studied the water sealed in a square silica microtubule under nuclear magnetic resonance (NMR) and reported its NMR H-spectrum width was much wider than that of the bulk water. They speculated that the movement of the water molecules in the 10–100 nm near-surface region of the enclosed microtubule exhibited slower translational motions and higher proton mobility of water. Ali Eftekhari-Bafrooei et al. [28,29,30,31] used vibrational sum frequency generation (vSFG) spectra to test the characteristic spectrum of water on the silica surface and found that the relaxation rate of OH stretching vibrational energy was significantly different from that of the bulk water. Teschke et al. [32] measured the dielectric constant distribution of water perpendicular to the mica surface using atomic force microscopy and found that the dielectric constant started to increase rapidly at about 10 nm from the surface. Ataka et al. [33] used surface-enhanced infrared spectroscopy to study the orientation of the water molecules at the interface between the Au(111) electrode surface and perchloric acid and found that the spectrum of the interfacial water changed dramatically in peak frequency and bandwidth from that of the bulk water. From this, they deduced a strongly hydrogen-bonded ice-like structure from the spectra.

The current research on the solid-surface-bound liquids has three distinct characteristics. First, most of the research methods use molecular simulation methods. Second, the fluid of study is mainly water. Third, the experimental observations of the spatial distribution of the solid–liquid interface were carried out without nanometre resolution. The properties and structural characteristics of the surface-bound water remain largely unexplored. Similarly, the anomalous behaviour of the surface-bound alkanes on the solid remains unknown.

Transmission electron microscopy (TEM) with sub-nanometre spatial resolution is an ideal tool for observing nanoscale structures at interfaces. However, the high vacuum (~10^−5^ Pa) operating environment of the TEM results in rapid evaporation of most liquids. Therefore, previous studies using TEM on liquids have been limited to using closed samples, where the internal pressure is maintained at atmospheric pressure [34,35,36]. Recently, Tomo et al. [37] reported the existence of super-stable ultra-thin water films in hydrophilized carbon nanotubes (CNTs) under high vacuum by TEM. For hexadecane [38] with lower saturated vapor pressure, it is currently unknown what its behaviour is like on the hydrophilic silica material at atmospheric or under high vacuum condition. This is an interesting and worthwhile question.

Research on this issue can promote the development of theories in many fields. For example, seepage in tight reservoirs is not a conventional linear seepage theory in petroleum development. Scholars use the boundary retention layer theory [7] to explain the nonlinear seepage phenomenon. However, there is currently no experimental evidence for a boundary retention layer. The research on the abnormal properties and structure of hexadecane on the surface of silica can promote the development of nonlinear seepage theory and propose a new technical direction for enhanced oil recovery of tight oil reservoirs.

In the paper, TEM and NMR spectroscopy were used to research the morphology and spectral characteristics of hexadecane in submicron to micron-scaled silica tubes at room temperature in this paper.

## 2. Methods

### 2.1. Method for Preparing TEM Samples

Silica microtubules purchased from PolyMicro Inc. (Phoenix, AZ, USA) were immersed in hydrofluoric acid (analytical grade, purchased from Shanghai Aladdin Bio-Chem Technology Co., Ltd., Shanghai, China) to dissolve away the polyimide coating and to thin it down, followed by washing in pure water and drying in a convection oven. Subsequently, the thinned-down microtubules were placed on a single-slit pure copper carrier net (purchased from Beijing Zhongjingkeyi Technology Co., Ltd., Beijing, China). Quick-drying glue was used to fix the microtubules perpendicular-wise to the slits on the carrier net followed by spraying with gold to enhance the conductivity of the sample. The treated microtubules fixed at the slits (Figure 1a and Appendix A) were processed by FIB-SEM (Helios NanoLab 600i, FEI, Hillsboro, OR, USA) to be made into TEM sections (TECNAI G2 F20, FEI, Hillsboro, OR, USA). Imaging of the microtubules in slits was performed using a dual-beam scanning electron microscope with an acceleration voltage of 2.5 kV and a current of 0.4 nA, in backscatter mode, with an in-column detector (ICD).

### 2.2. NMR Sample Preparation Method

The microtubules were cut into 14 cm long sections, followed by immersing ten of these so-treated samples in hydrofluoric acid to strip its coating and cleansing them afterwards with purified water (manufactured from Nanopure Diamond UV/UF, Thermo Fisher Scientific, Waltham, MA, USA). Then the samples were dried in an oven and filled with hexadecane (80 microtubules 0.2 μm in ID, were used for testing with hexadecane and 10 empty microtubules 0.2 μm in ID were used as control). The H-spectrum [27] of the hexadecane (analytical grade, purchased from Shanghai Hansi Chemical Co., Ltd., Shanghai, China) in the microtubules was measured using a nuclear magnetic resonance spectrometer (AVANCE NEO, Bruker, Billerica, MA, USA). All NMR spectra were measured with a Bruker AVANCE NEO spectrometer at 700 MHz without spinning and locking. The scanning range is −1~13 ppm for 1H.

### 2.3. Preparation Method to Adjust the Silanol Density of the Sample

The microtubules were soaked in hydrofluoric acid to peel off the coating followed by cleaning with pure water. A high-pressure plunger pump (Teledyne Isco D, Teledyne Isco, Lincoln, NE, USA) was used to inject the treated microtubules with saturated sodium hydroxide solution. After letting it stand in sodium hydroxide for 8 h, the microtubules were cleaned with injected DI water followed by drying in a convection oven.

After making the TEM samples, submicron holes were milled into the silica slices using an ion beam (Figure 1b). The samples were soaked in hexadecane for 1 min and left exposed in the atmosphere for a period of either 2 or 8 weeks, followed by observation under transmission electron microscope. A FEI Tecnai G2 F20 microscope [37] with variable electron voltage (120 and 200 keV), camera Gatan US 1000XP-P, detector EDX Oxford Instruments XMAX, at a magnification of 490,000 was used.

## 3. Results

### 3.1. The Discovery of Anomalous Wall-Bound Hexadecane in the Silica Microtubules

The microtubules 5 μm in ID were cut longitudinally along the axis using an ion beam on a double-beam scanning electron microscope, followed by further cutting it into thin slices with a thickness of 100 nm. Figure 2a shows a TEM image of the resultant section of the treated microtubule. It can be clearly seen from Figure 2a that the matrix wall of the silica microtubule and the interior content of the microtubule have a clear interface showing that there is no substance on the interior. In contrast, Figure 2b shows the result of the microtubule slices after soaking in hexadecane and put under TEM for observation. The figure clearly shows that there is a dendritic layer of material growth with thickness of about 100 nm between the matrix wall and the vacuumed centre of the microtubule. This layer of material only appears on the inner tube wall, and does not exist on the freshly sliced surface. This layer of material can exist stably in a high vacuum environment at room temperature, indicating that the material is non-volatile in nature and can withstand high continuous vacuum.

Silica microtubular sections of different ID were soaked in hexadecane and then taken out for TEM observation. Figure 3a–d shows the TEM images. It can be seen from the Figure 3a–d, that the interfaces between the wall-bound layer and the vacuum in microtubule 15 μm and 30 μm in ID are dendritic, with fewer dendritic structures for the latter. However, the interfaces of the wall-bound layer and the vacuum for the microtubule 2 μm and 0.2 μm in ID are not dendritic, but wavy. Figure 3e shows the plot of the thickness of the wall-bound layer vs the ID of the microtubules. It can be seen from Figure 3e that the thickness of the wall-bound layer of the microtubule increases sharply with the decrease in microtubular ID. The thickness of the wall-bound layer starts at 12 nm for ID of 30 μm, increases to 98 nm for ID of 5 μm, follows by a further rapid increase to 400 nm for microtubular ID of 0.2 μm. This wall-bound layer appears after the sample is soaked in hexadecane, but whether this layer is hexadecane requires further experimental validation.

The hexadecane was injected into silica microtubule 0.4 μm in ID at room temperature, and an ion beam was used to cut the microtubule longitudinally along the axis to prepare the TEM samples. Figure 4 shows the longitudinal cross-sectional profile of a hexadecane slice shown centred horizontally across. The content of the microtubule is shown in the middle. The bright white areas in the centre and the left are the hollow spaces, while the grey areas in the central-right of the image are the hexadecane slices. There is a clear meniscus between the hollow space and the hexadecane (Energy spectrum data in Appendix A Appendix A).

It can be clearly seen from Figure 4 that a stable non-volatile hexadecane is still present in the sectional cut from the 0.4 μm microtubule. Compared to the hard silica matrix wall with which it is bound to, this medium can equally withstand the focused milling of an ion beam to be shaped into slices of thickness of only 0.1 μm. Moreover, such a thin slice appears to be very stable while being wholly exposed to the environment at 20 °C, high vacuum of 10^−5^ Pa, and still maintain its shape for a long time. From these observations, this wall-bound layer appears to be a special kind of solid.

The energy spectrum of the hexadecane in silica microtubule 2 μm in ID are shown in Figure 5 (Data for microtubule 5 µm in ID are shown in Appendix A Appendix A). Figure 5b shows that with the exception of the three peaks for carbon, silicon and oxygen in the energy spectrum corresponding to the elemental makeup of silica microtubule and the hexadecane content within the microtubule, there is no other elements besides a very small trace amount of copper coming from the copper mesh. Figure 5c–f depict the distribution of carbon, silicon and oxygen in Figure 5a, respectively. Silicon has the same distribution as oxygen, characterizing the walls of the silica on one side of the picture. In the near-wall region within the silica microtubule, carbon elements gather here and extend to a certain thickness outside the glass matrix. There is only carbon in this layer, which proves that this layer was indeed hexadecane and this abnormal hexadecane was not affected by any other impurities.

### 3.2. Abnormal Structural Characteristics of the Hexadecane on the Inner Surface of Silica Microtubules

The electron diffraction test of the hexadecane on the inner surface of the silica microtubules by TEM (Figure 6a) shows that its electron diffraction spectrum is neither symmetrically shaped typical of a single crystalline structure nor that of a diffraction ring centred on the transmission point typical of a polycrystalline-structure, but rather that of a diffuse halo similar to that of an amorphous structure.

The hexadecane in the microtubule was tested at room temperature by nuclear magnetic resonance H-spectrum. The characteristic peak of hexadecane in the silica microtubule 15 μm and 5 μm in ID was found to be a doublet of 1.26 ppm and 0.882 ppm. This is the NMR H-spectrum characteristic peak of the conventional hexadecane. Comparing it to that of the hexadecane in microtubule 0.2 μm in ID, the characteristic peak is shifted to the high field by about 0.16 ppm and the characteristic peak width is much wider and is similar to that of the solid. This indicates that the hexadecane molecules in the silica microtubule 200 nm in ID moves more slowly and has a higher viscosity than the bulk hexadecane molecules.

### 3.3. The High-Density Aggregation of Silanol on the Silica Microtubule Interior Is an Important Reason for the Formation of Anomalous Hexadecane

The H-spectrum data of the silica microtubule of five different ID sizes were measured by nuclear magnetic resonance. In Figure 7a, there are two absorption intensity peaks at 3.4 and 1.5 ppm for silica microtubules with different inner diameters. According to the literature reports [39,40,41], these two peaks in the H spectrum are characteristic peaks of the silanol on the silica surface with their peak integration characterizing the silanol content. From Figure 7a, it can be seen that the content of silanol in the microtubule material composition sharply increases with the decrease in the microtubular ID. The distribution of the silanol can be obtained by measuring the amount of silanol in the microtubular matrix material before and after the water injection by H-NMR. After water injection, the hydrolysis reaction of the silanol on the interior of the microtubule resulted in the presence of the H element in the silanol in the form of the hydronium ions in water. The peak height of silanol decreased by 99% in Figure 7b. This indicates that most of the silanol in the microtubules are distributed on the inner surface of the microtubules.

From the integral value of the silanol characteristic peak and the microtubular diameter, the silanol density at the inner wall of the microtubules can be obtained. Combined with the thickness of the anomalous hexadecane at the wall in Figure 3, Figure 7c is obtained. It can be seen from Figure 7c that there is a good positive correlation between the thickness of the anomalous hexadecane and the logarithm of the relative density of the silanol. With the increase in silanol density, the thickness of the hexadecane on the inner wall of the microtubules also increased rapidly. This suggests that the surface silanol density played an important role in the formation of anomalous alkanes.

In order to further confirm the effect of surface silanol on the formation of the anomalous wall-bound hexadecane layer, further testing was conducted varying the silanol density on the inner surface of the silica microtubules.

Pores 300 nm in ID were fabricated using ion beams on the synthesized TEM sections of the silica microtubules. Two of thus-treated samples were placed at room condition for a period of 2 and 8 weeks, respectively. After soaking it in the hexadecane, the interior surface of the submicron pores was put under observation under a TEM. It can be seen from the sample placed in hexadecane for 2 weeks in Figure 8a, that most of the inner edge of the hole has a clear and obvious interface with the hexadecane, with only a few places exhibiting dendritic interface with the hexadecane dendrites having a maximum height of only 20 nm. It can be seen from the Figure 8b for the sample placed in hexadecane for 8 weeks (Repeatability experimental results in Appendix A Appendix A), the dendritic hexadecane was significantly denser than the sample at 2 weeks, with most of the dendrites more than 50 nm in height. Most of the pore space with a diameter of about 300 nm are occupied by the hexadecane, and the maximum thickness of the hexadecane on the inner wall of the submicron hole reaches 130 nm.

Although silica is chemically stable, the silica surface is susceptible to hydrolysis with atmospheric water vapour. Surface silanol can be produced by hydroxylation of the siloxane groups. The silanol groups are the source of the hydrophilic character of silica, and their concentration can increase with time [42,43,44,45]. Wunder et al. [46,47,48] believed that the density of silanol groups on the surface of silica is related to the size of the pore. The smaller the submicron nanopores, the larger the surface defects, and the greater the density of the silanol groups generated from reaction with the water vapor in the air after being exposed to the open atmosphere for a period of time. This can explain the increase in anomalous hexadecane on the pore walls of the submicron silica with the increase in time. With the increase in time, the density of the silanol at the interior wall of the submicron pores increases continuously. The greater the density of silanol, the thicker the resultant anomalous wall-bound hexadecane.

A saturated NaOH solution was used to lower the silanol surface density on the interior surface of the silica microtubule 2 μm in ID due to the reaction of saturated NaOH solution with silica to form a stable silicate. Figure 8c shows the NMR H-spectrum of the microtubule treated with saturated NaOH solution. It can be seen from the relative size of the characteristic peaks of the silanol at 3.4 and 1.5 ppm, the NMR H-spectrum characteristic peaks of the untreated microtubule are much lower than that of the original untreated microtubule. The integrated area of the NMR characteristic peaks of the treated microtubules is only 2% of that of the untreated microtubules. This indicated that the silanols on the inner wall of the microtubules were basically eliminated after the saturated NaOH solution treatment. The microtubules thus-treated with the saturated NaOH solution were made into TEM sections. After soaking in hexadecane, the samples were put under observation by TEM. The results are shown in Figure 8d. From the image of the surface of the silica, the maximum height of the dendritic hexadecane is observed to be only 20 nm. Additionally, Dendritic anomalous hexadecane is sparse. In contrast to the results shown in Figure 3b, the hexadecane at the inner wall of the microtubule 2 μm in ID occupies an entire layer on the interior wall surface and has a thickness of up to 241 μm. This shows that the abnormal hexadecane on the inner wall of the microtubules treated with saturated NaOH solution is greatly reduced.

## 4. Discussion

This research is aimed at characterizing the changes in properties of the hexadecane in silica microtubules. It was found that a non-volatile hexadecane existed under normal temperature and high vacuum conditions on the inner surface of the silica microtubule after contacting with hexadecane. TEM is used as the primary tool to characterize the hexadecane on the silica surface through observation of the nanoscale solid–liquid interfaces. However, it must be emphasized that the observation in this study were not influenced by artifacts [49,50] or impurities caused by the sample preparation process. Comparing the image of the microtubule slice before and after soaking in hexadecane, it can be seen that the interface between the hexadecane and the exhibits an irregular interface, which is finger-like [51]. A typical meniscus between the hexadecane in the microtubule and the vacuum can be seen in the slice of the hexadecane in the microtubule 0.4 micron in ID under the TEM. Both the finger-like interface and the presence of a meniscus are typical morphological features of a fluid. This suggests the observed material inside the silica microtubule is a fluid. The areal scan data of the energy spectrum show that in the range of approximately 100 nm from the surface of the silicon dioxide, there are no elements of silicon and oxygen, but only carbon. Thus, this suggests that the 100 nm thick carbon element on the interior silica microtubular wall can only come from hexadecane instead of impurities.

According to the phase behaviour of conventional hexadecane, it should be a gas under normal temperature and high vacuum conditions. According to the Kelvin equation [52], the curved liquid–gas interface in microtubules causes changes in the vapour pressure of the liquid. Although the saturated vapor pressure of the hexadecane is extremely low [38], the vapour pressure of the alkane is still much lower than that of the high vacuum environment, even for hexadecane in microtubules 200 nm in ID. Thus, the Kelvin equation does not explain the stable existence of hexadecane in high vacuum. In the microtubules with ID >2 μm, the abnormal hexadecane only appeared on the near-wall region of the microtubule, which indicated that the condition necessary for the formation of the abnormal hexadecane might be related to the scale-induced surface properties. Through the use of nuclear magnetic resonance, it was found that a large number of silanol groups were accumulated on the inner wall of the microtubule of ID <2 μm. The density of the silanols increases rapidly with decreasing microtubular radius. The surface density of the silanol in silica microtubules submicron in ID is 10^7^ times that in silica microtubule 30 μm in ID. There is a good positive correlation between the silanol surface density and the thickness of the anomalous hexadecane in microtubules of different ID. This suggests a clear relationship between the surface silanol density and the thickness of the anomalous hexadecane layer. The experiment on the effect of silanol groups on the surface of the silica on the abnormal hexadecane formation indicated that the density of the silanol groups on the silica surface played a very important role in the formation of the anomalous hexadecane. This corroborates the finding of Tomo et al. [37] which reported presence of ultra-stable water in the vacuum environment of hydrophilic carbon nanotubes. The greater the silanol density on the silica surface, the more hydrophilic the silica surface [53], and the thicker the abnormal hexadecane layer on the surface. As additional support, Argyris et al. [15] used molecular simulation experiments and determined that there are two distinct water layers formed on the surface of the silica covered with high density of hydroxyl groups. Sanders et al. [54] reported a similar change in the interfacial water structure after tuning the OH groups on the silica surface. Urashima et al. [55] reported the presence of water molecules gathering around the silanol group for the formation of strongly hydrogen-bonded droplets.

In this paper, the discovery of solid-like hexadecane on the surface of silica microtubule not only helps to theoretically explain some anomalous physical phenomena, but also has broad application prospects. For example, this finding can possibly explain the nonlinear seepage phenomenon of single-phase liquid alkanes in the submicron to nanometre sandstone pore throat in the field of enhanced oil recovery [56,57]. This finding can also be applicable as a novel non-frozen sample preparation method for biological sample preparation in life sciences and other fields; as a novel micro-nanofluidic control in the field of micromechanics; micro-nano catalysts in the field of chemical synthesis; etc.

In this paper, the discovery of the unusually stable solid-like hexadecane found in the silica microtubules posed a series of additional scientific questions that need to be addressed. For example, what is the molecular structure of this unusual hexadecane? Besides hexadecane, what are other possible media (such as liquid alkanes, ethanol, and water) which can also turn solid-like in the enclosed spaces on the micron to submicron scale? What is the intrinsic mechanism leading to the formation of the solid-like hexadecane anomalous structure, etc.?

## 5. Conclusions

This research centred on answering the questions of whether hexadecane can exist stably in silica microtubule under high vacuum conditions and discussed the finding of the main influencing factors. In this paper, the morphology the hexadecane on the inner surface of the silica microtubules was put under observation by transmission electron microscopy, and the characteristic structure of the hexadecane in the microtubules was analysed by NMR H-spectrum. By examination under TEM, it was found that there was a layer of non-volatile wall-bound hexadecane in the silica microtubules after soaking in hexadecane. The thickness of the non-volatile hexadecane increases as the ID of the microtubule decreases, from 12 nm in microtubule 30 μm in ID to 400 nm in microtubule 0.2 μm in ID. The hexadecane in microtubule 0.4 μm in ID could be milled like a solid into thin slices of about 100 nm. The hexadecane in microtubule 0.2 μm in ID was characterized under H-NMR. Comparing the NMR H-spectra of hexadecane in microtubule 30 μm in ID, it was found that the characteristic peak of non-volatile hexadecane shifted to the high field by 0.16 ppm, and the characteristic peak width became wider, similar to that of the solid. It was found that the silanol density of the inner wall of the silica microtubule increased rapidly with the decrease in the microtubular diameter by H-NMR, which was consistent with the change in the thickness of non-volatile hexadecane layer with the microtubular diameter. With the increase in time, the thickness of the abnormal wall-bound hexadecane on the interior surface of the newly shaped submicron pores of the silica microtubules increased continuously through TEM observation. Saturated sodium hydroxide was used to reduce the silanol groups on the interior silica surface. After soaking the NaOH-treated silica microtubule in hexadecane, the content of the anomalous hexadecane in the silica microtubule 2 μm in ID was greatly reduced as compared to the control. These experimental results show that the concentration of silanols on the interior surface is an important determinant for the formation of the anomalous wall-bound hexadecane layer within the silica microtubule. Together with the finding of previous reports of a solid-like liquid found on the solid surface, the findings of this study suggest that the hexadecane in the silica microtubule with high interior silanol surface density is a solid-like substance which can remain non-volatile under high vacuum. This property of the hexadecane has wide potential application value. Further investigations to examine the intrinsic mechanism of the interaction between surface silanol density and its effect on other media is warranted.

## Figures and Tables

**Figure 1 materials-16-00009-f001:**
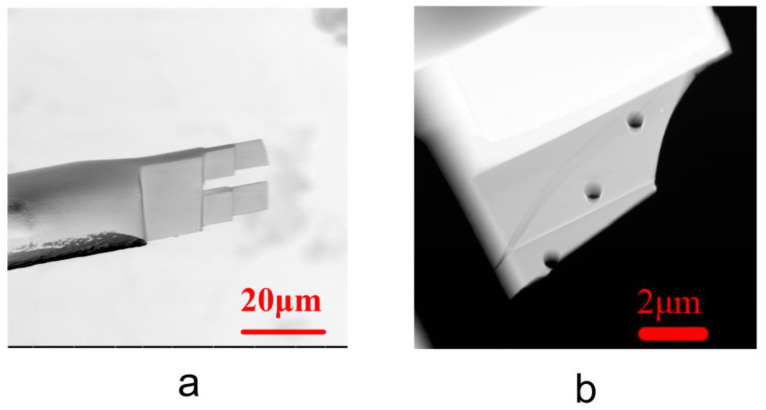
Photographs of microtubule sections and submicron pores on the sections. (**a**) Scanning electron micrographs of TEM sections made along the axial direction of the microtubules using an ion beam. (**b**) Submicron processed using an ion beam perpendicular to the TEM section of the TEM image.

**Figure 2 materials-16-00009-f002:**
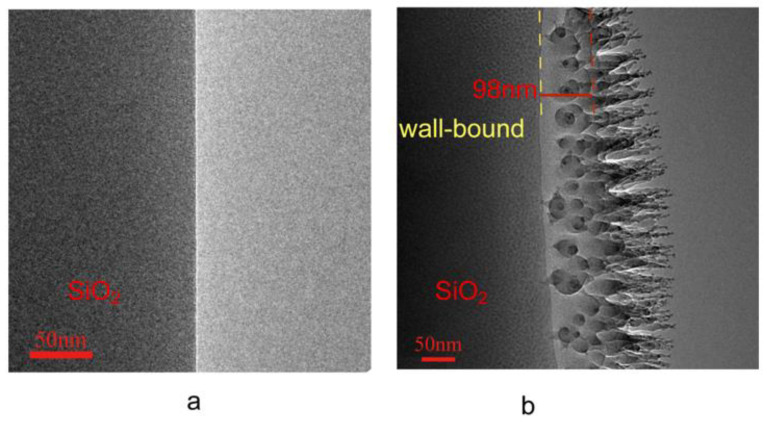
Sectional TEM image of sliced samples in silica microtubules 5 μm in ID before and after soaking in hexadecane. (**a**) is the TEM image before soaking in hexadecane. (**b**) is the TEM image after soaking in hexadecane.

**Figure 3 materials-16-00009-f003:**
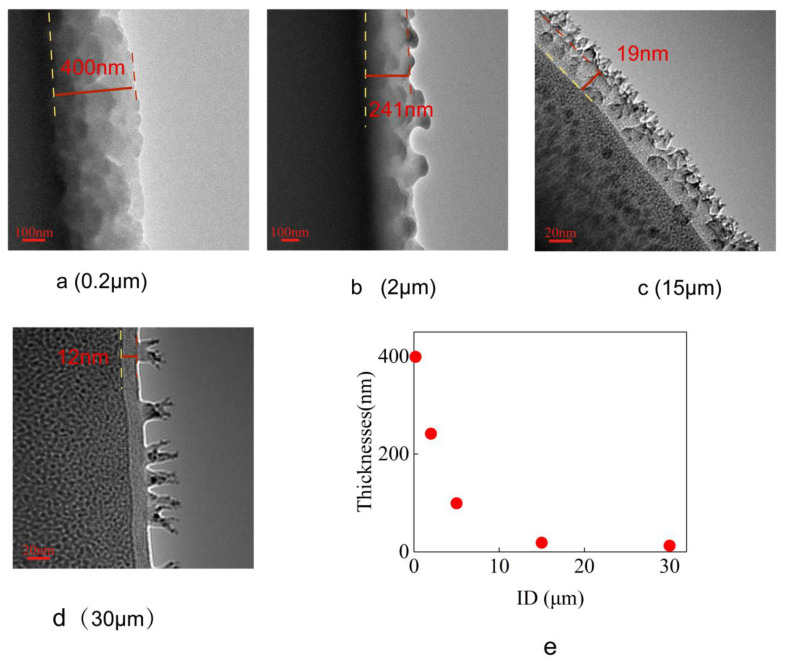
Sectional TEM image of samples in silica microtubules 0.2 μm, 2 μm, 15 μm and 30 μm in ID and the relationship between thickness of the wall-bound hexadecane layer and the microtubular ID. (**a**–**d**) Cross-sectional TEM images of samples with microtubule 0.2 μm, 2 μm, 15 μm and 30 μm in ID, respectively. (**e**) The relationship between the wall-bound hexadecane layer’s thickness and the microtubule’s inner diameter. The horizontal coordinate is the inner diameter of the microtubule, and the vertical coordinate is the thickness of the wall-bound hexadecane layer.

**Figure 4 materials-16-00009-f004:**
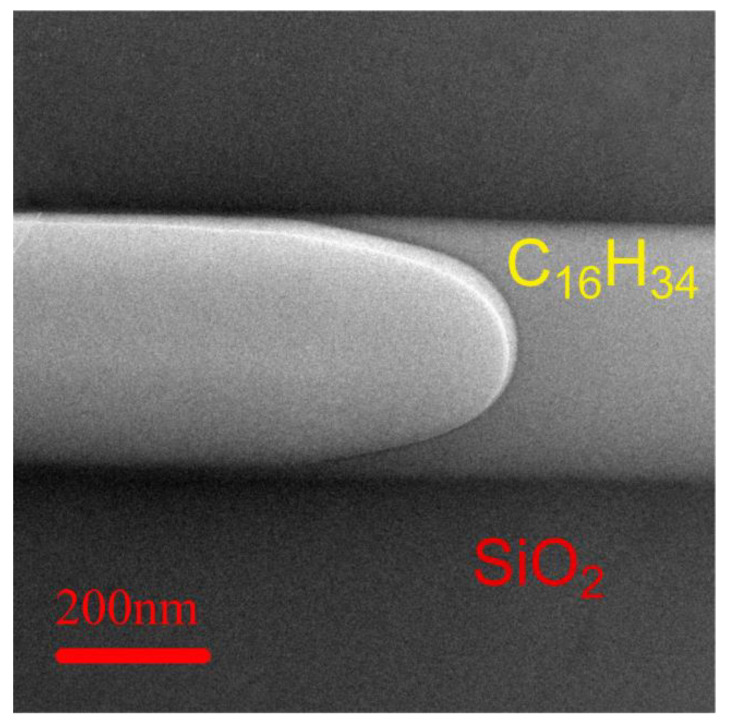
Sectional TEM image of samples in silica microtubules 0.4 μm in ID.

**Figure 5 materials-16-00009-f005:**
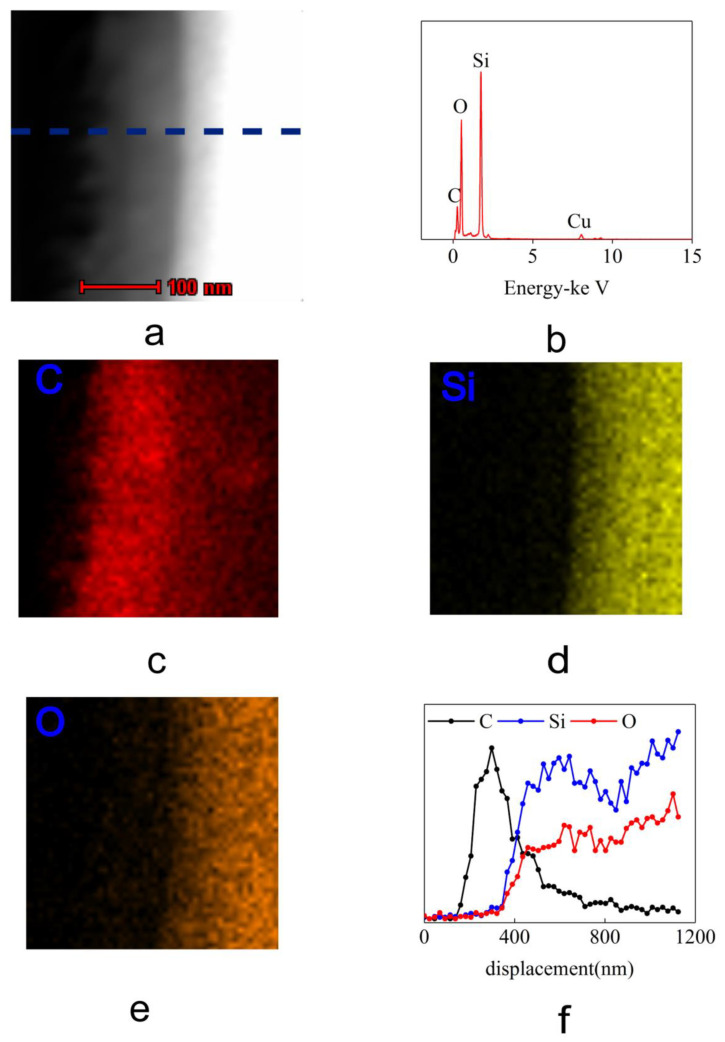
Energy Dispersive Spectroscopy Data of hexadecane on the inner surface of silica microtubules (ID of 2 μm). (**a**) is the high-angle annular dark-field (HAADF) imaging of the near-wall region of the microtubule after infiltration with hexadecane. (**b**–**e**) are the energy spectrum data and the EDS areal scans of the C, Si, O elements taken of the same area as in (**a**). (**f**) is the EDS line scans of the C, Si, O elements taken of the same area as the blue lines in (**a**).

**Figure 6 materials-16-00009-f006:**
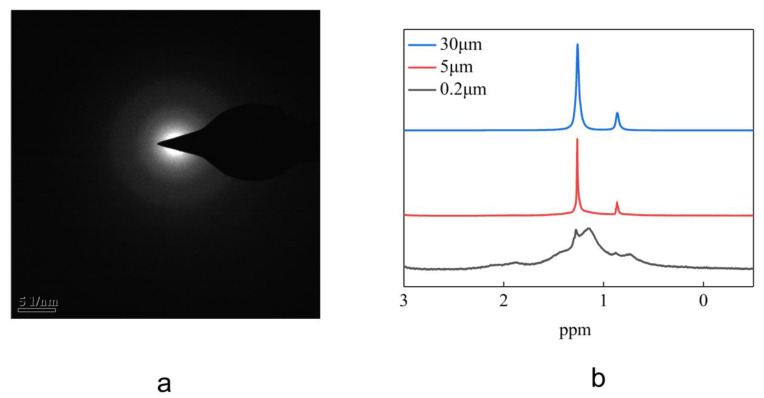
Electron diffraction patterns and NMR H-spectrum of the hexadecane in the silica microtubules. (**a**) shows the electron diffraction patterns of the hexadecane in silica microtubules 2 μm in ID. (**b**) depicts the H-NMR spectrum of the hexadecane in silica microtubules 30 μm, 5 μm and 0.2 μm in ID.

**Figure 7 materials-16-00009-f007:**
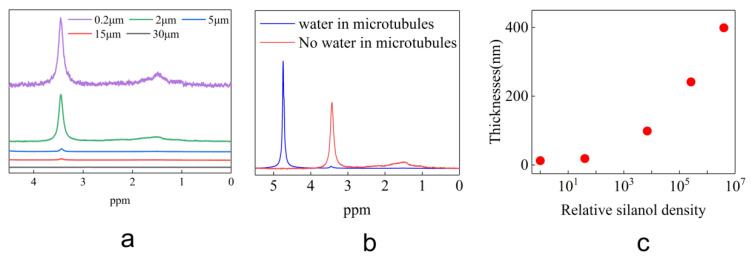
Distribution of Silanol Content on Microtubule Surface. (**a**) shows the H-NMR spectral data of the silica microtubule of different ID sizes for which the peak area characterizes the quantities of the silanol groups. (**b**) shows the H-NMR spectral data of empty silica microtubules and water-filled microtubules. (**c**) shows the relationship between the relative silanol density and the thickness of wall-bound water layer and the signal intensity of the H-spectrum for microtubules with different ID sizes based on the silanol density found within glass microtubule 30 μm in ID.

**Figure 8 materials-16-00009-f008:**
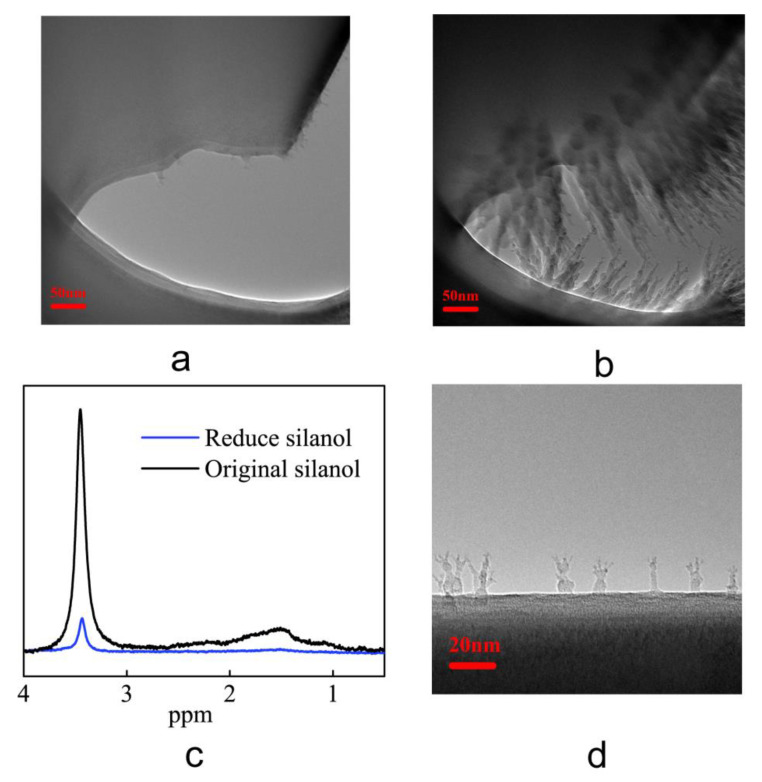
Effect of surface silanol content on the formation of the anomalous hexadecane. (**a**,**b**) The transmission electron micrographs of the samples placed for 2 weeks and 8 weeks after exposure to hexadecane. (**c**) The NMR H spectra of microtubules before and after treatment with saturated NaOH solution. (**d**) The transmission electron micrographs of the inner wall of microtubules treated with saturated NaOH solution.

## Data Availability

Not applicable.

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
