# Peer review of "A Form of Non-Volatile Solid-like Hexadecane Found in Micron-Scale Silica Microtubule"

_materials, 2022, doi:10.3390/ma16010009_

Round 1
Reviewer 1 Report
Review comments attached.

Author Response
Dear Reviewers:
Thank you for your letter and the comments concerning our manuscript entitled "A form of non-volatile solid-like hexadecane found in micron-scale silica microtubule" (ID∶2037289). Those comments are all valuable and very helpful for revising and improving our paper, as well as the essential guiding significance to our research. We have studied the comments carefully and made a correction that we hope meets with approval. Revised portions are marked in red on the paper. The leading corrections in the paper and the responses to the reviewer's comments are as follows:
Responds to the reviewer's comments:
Reviewer #1:
- Response to comment:(Methods: More experimental details should be added. Brands/sources of chemicals and operating conditions of instruments (e.g. voltages of electron microscope beams) are typically included in research reports. )
Response∶We have added more experimental details. Made the following additions
- hydrofluoric acid (analytical grade,purchased from Shanghai Aladdin Bio-Chem Technology Co., LTD).
- pure copper carrier net(purchased from Beijing Zhongjingkeyi Technology Co., Ltd)
- the hexadecane (analytical grade,purchased from Shanghai Hansi Chemical Co., Ltd)
- transmission electron microscope(200KV FEI Tecnai G2 F20).
- Imaging of the microtubules in slits was performed using a dual-beam scanning electron microscope (FEI Helios NanoLab 600i) with an acceleration voltage of 2.5 kV and a current of 0.4 nA, in backscatter mode, with an in-column detector (ICD).
- All NMR spectra were measured with a Bruker AVANCE NEO spectrometer at 700MHz without spinning and locking. The scanning range is -1~13ppm for 1H.
- A FEI Tecnai G2 F20 microscope [36] with variable electron voltage (120 and 200 keV), camera Gatan US 1000XP-P, detector EDX Oxford Instruments XMAX, at a magnification of 490,000 was used.
2.Response to comment:(Figure 8: The caption is incomplete. List exactly what is illustrated by each square labeled a, b, c, and d.)
Response∶We modified the callout for Figure 8. The modified results are as follows
Figure 8. effect of surface silanol content on the formation of the anomalous hexadecane.Figure 8(a, b) shows the transmission electron micrographs of the samples placed for 2 weeks and 8 weeks after exposure to hexadecane. Figure 8(c) shows the NMR H spectra of microtubules before and after treatment with saturated NaOH solution.Figure 8(d) shows the transmission electron micrographs of the inner wall of microtubules treated with saturated NaOH solution.
3.Response to comment:(How does saturated NaOH eliminate silanols on microtubule surfaces? Strong base is known to hydrolyze and dissolve silica. Does NaOH under this condition deprotonate surface silanols and promote siloxane bond formation? A one sentence explanation would be a little helpful. The end result is not in dispute; the NMR evidence is conclusive.)
Response∶Silica reacts with saturated sodium hydroxide to form a stable silicate. Silicate covers the silica. Due to the decrease in silica, the density of silanols also decreases.
We have added the text's reason for the silanol density decrease. The modified results are shown below.
A saturated NaOH solution was used to lower the silanol surface density on the interior surface of the silica microtubule 2μm in ID due to the reaction of saturated NaOH solution with silica to form a stable silicate.
Reviewer 2 Report
Dear Authors,
I have carefully read the submitted manuscript. There are the following remarks:
1. In the review part, it is necessary to describe in more detail for what practical purposes the effect under study is used and what areas of technology its study can affect (specifically).
2. At the end of the introduction, the purpose of the study for this work should be clearly stated. At the moment it looks like research for the sake of research.
3. The operating modes of SEM, TEM and NMR should be clearly described in the second section so that, if possible, the reader can reproduce these experiments.
4. In conclusion, it is necessary to detail exactly what practical significance the revealed effect at the solid-liquid phase boundary has, and how exactly it is supposed to be used to achieve specific scientific and technical results.
Author Response
Dear Reviewers:
Thank you for your letter and the comments concerning our manuscript entitled "A form of non-volatile solid-like hexadecane found in micron-scale silica microtubule" (ID∶2037289). Those comments are all valuable and very helpful for revising and improving our paper, as well as the essential guiding significance to our research. We have studied the comments carefully and made a correction that we hope meets with approval. Revised portions are marked in red on the paper. The leading corrections in the paper and the responses to the reviewer's comments are as follows:
Responds to the reviewer's comments:
Reviewer #2:
- Response to comment:(In the review part, it is necessary to describe in more detail for what practical purposes the effect under study is used and what areas of technology its study can affect (specifically). )
Response∶We have added the actual application field. Made the following additions
Liquids on solid surfaces are involved in many surface reactions. These surface reactions include electrochemistry at liquid/solid interfaces,[1] heterogeneous catalyses such as hydrogen production by catalytic reforming, [2] rock weathering, and corrosion of metals[3]. The properties of the liquid at the solid interface also significantly impact the liquid's heat transfer[4] and flow characteristics on the solid surface, especially in small-scale spaces—for example, micro-nano-scale pipe flow [5,6] and nonlinear seepage [7] in reservoirs and other issues.Despite the importance of solid-liquid interfacial science, much anomalous behaviour of the liquids on solid surfaces is still far from being thoroughly researched, with many new phenomena yet to be discovered and understood.
2.Response to comment:(At the end of the introduction, the purpose of the study for this work should be clearly stated. At the moment it looks like research for the sake of research.)
Response∶We have added a paragraph at the end of the Introduction to explain the significance of this study in the petroleum development field and our study's purpose.
. The added paragraph is as follows.
Research on this issue can promote the development of theories in many fields. For example, seepage in tight reservoirs is not a conventional linear seepage theory in petroleum development. Scholars use the boundary retention layer theory to explain the nonlinear seepage phenomenon. However, there is currently no experimental evidence for a boundary retention layer. The research on the abnormal properties and structure of hexadecane on the surface of silica can promote the development of nonlinear seepage theory and propose a new technical direction for enhanced oil recovery of tight oil reservoirs.
3.Response to comment:(The operating modes of SEM, TEM and NMR should be clearly described in the second section so that, if possible, the reader can reproduce these experiments.)
Response∶We added SEM, TEM and NMR modes of operation in the second section. The added part is as follows.
Part FIB-SEM
Imaging of the microtubules in slits was performed using a dual-beam scanning electron microscope (FEI Helios NanoLab 600i) with an acceleration voltage of 2.5 kV and a current of 0.4 nA, in backscatter mode, with an in-column detector (ICD).
Part H-NMR
All NMR spectra were measured with a Bruker AVANCE NEO spectrometer at 700MHz without spinning and locking. The scanning range is -1~13ppm for 1H.
Part TEM
A FEI Tecnai G2 F20 microscope with variable electron voltage (120 and 200 keV), camera Gatan US 1000XP-P, detector EDX Oxford Instruments XMAX, at a magnification of 490,000 was used.
4.Response to comment:(In conclusion, it is necessary to detail exactly what practical significance the revealed effect at the solid-liquid phase boundary has, and how exactly it is supposed to be used to achieve specific scientific and technical results.
)
Response∶We have added a paragraph at the end of the Introduction to explain the significance of this study in the petroleum development field
Research on this issue can promote the development of theories in many fields. For example, seepage in tight reservoirs is not a conventional linear seepage theory in petroleum development. Scholars use the boundary retention layer theory to explain the nonlinear seepage phenomenon. However, there is currently no experimental evidence for a boundary retention layer. The research on the abnormal properties and structure of hexadecane on the surface of silica can promote the development of nonlinear seepage theory and propose a new technical direction for enhanced oil recovery of tight oil reservoirs.
Reviewer 3 Report
The manuscript entitled: “A form of non-volatile solid-like hexadecane found in micron-2 scale silica microtubule” is a very interesting and well written paper. The paper deserves publication after the following revisions:
L. 19-23: Authors must refer in past tense for already done issues.
L. 24-25: Must be rewritten. It does not make sense.
L. 32: Regarding electrochemistry, more references ought to be taken into account, such as https://doi.org/10.30955/gnj.000770, http://dx.doi.org/10.5004/dwt.2022.28133
Please pay attention on the unnecessary spaces and the use of punctuation in the whole manuscript (indicative deficiencies: l.42, 139, 143, 144, 148, 155, 179, 188, 205 and many others). Carefully read the entire manuscript and repair.
Sections 2.1, 2.2 and 2.3: The references of the preparation method protocols are missing.
Result: Use it in plural form.
Conclusions: Do not use references in the conclusion section. It is not common; especially when you have earlier referred to them.
Author Response
Dear Reviewers:
Thank you for your letter and the comments concerning our manuscript entitled "A form of non-volatile solid-like hexadecane found in micron-scale silica microtubule" (ID∶2037289). Those comments are all valuable and very helpful for revising and improving our paper, as well as the essential guiding significance to our research. We have studied the comments carefully and made a correction that we hope meets with approval. Revised portions are marked in red on the paper. The leading corrections in the paper and the responses to the reviewer's comments are as follows:
Responds to the reviewer's comments:
Reviewer #3:
- Response to comment:( 19-23: Authors must refer in past tense for already done issues.L. 24-25: Must be rewritten. It does not make sense. )
Response∶We have modified the tense and rewritten the sentences in L. 24-25. The modified result is shown below.
the characteristic peaks of this abnormal hexadecane were shifted to the high field with a broader characteristic peak, nuclear magnetic resonance hydrogen spectroscopy spectral features typical of that of solids. The surface density of these abnormal hexadecanes was found to be positively correlated with the silanol groups found on the interior silica microtubular surface. This positive correlation indicates that the high-density aggregation of silanol is an essential factor for forming the abnormal hexadecane reported in this paper.
2.Response to comment:(L. 32: Regarding electrochemistry, more references ought to be taken into account, such as https://doi.org/10.30955/gnj.000770, http://dx.doi.org/10.5004/dwt.2022.28133)
Response∶We have added more references to electrochemistry. The result of the addition is shown below.
Liquids on solid surfaces are involved in many surface reactions. These surface reactions include electrochemistry at liquid/solid interfaces,[1-3] heterogeneous catalyses such as hydrogen production by catalytic reforming, [4] rock weathering, and corrosion of metals[5].
[2] Dermentzis K , Christoforidis A , Valsamidou E . Removal of hexavalent chromium from electroplating wastewater by electrocoagulation with iron electrodes. Global Nest Journal. 13(2011) 412-418.
[3] Marmanis D, Diamantis V, Thysiadou A, et al. Comparison of electro-oxidation (BDD anode and Ti/Pt cathode) and electro-coagulation (aluminum electrodes) for the treatment of raw landfll leachate. Desalination and Water Treatment. 260(2022) 203-208.
3.Response to comment:(Please pay attention on the unnecessary spaces and the use of punctuation in the whole manuscript (indicative deficiencies: l.42, 139, 143, 144, 148, 155, 179, 188, 205 and many others). Carefully read the entire manuscript and repair.)
Response∶We have searched and edited the full text.
4.Response to comment:(Sections 2.1, 2.2 and 2.3: The references of the preparation method protocols are missing. )
Response∶We have added references.The result is as follows
The H-spectrum [26] of the hexadecane (analytical grade,purchased from Shanghai Hansi Chemical Co., Ltd) in the microtubules was measured using a nuclear magnetic resonance spectrometer
A FEI Tecnai G2 F20 microscope [36] with variable electron voltage (120 and 200 keV), camera Gatan US 1000XP-P, detector EDX Oxford Instruments XMAX, at a magnification of 490,000 was used.
5.Response to comment:(Result: Use it in plural form.Conclusions: Do not use references in the conclusion section. It is not common; especially when you have earlier referred to them.)
Response∶We added plurals and removed references.The result is as follows
Results